# Surface Integrity Investigation to Determine Rough Milling Effects for Assessment of Machining Allowance for Subsequent Finish Milling of Alloy 718

**Jonas Holmberg [1],\*** **, Anders Wretland [2], Johan Berglund [1], Tomas Beno [3] and Anton Milesic Karlsson [4]**

[1]   RISE, IVF AB, Manufacturing, Argongatan 30, 431 53 Mölndal, Sweden; johan.berglund@ri.se
[2]   GKN Aerospace Sweden AB, 461 38 Trollhättan, Sweden; anders.wretland@gknaerospace.com
[3]   Department of Engineering Science, University West, 461 86 Trollhättan, Sweden; tomas.beno@hv.se
[4]   Tooltec Trestad AB, 461 38 Trollhättan, Sweden; anton.karlsson@tooltec.se
\*   Correspondence: jonas.holmberg@ri.se; Tel.: +46-70-780-60-72

**Abstract:** The planned material volume to be removed from a blank to create the final shape of a part is commonly referred to as allowance. Determination of machining allowance is essential and has a great impact on productivity. The objective of the present work is to use a case study to investigate how a prior rough milling operation affects the finish machined surface and, after that, to use this knowledge to design a methodology for how to assess the machining allowance for subsequent milling operations based on residual stresses. Subsequent milling operations were performed to study the final surface integrity across a milled slot. This was done by rough ceramic milling followed by finish milling in seven subsequent steps. The results show that the up-, centre and down-milling induce different stresses and impact depths. Employing the developed methodology, the depth where the directional influence of the milling process diminishes has been shown to be a suitable minimum limit for the allowance. At this depth, the plastic flow causing severe deformation is not present anymore. It was shown that the centre of the milled slot has the deepest impact depth of 500 μm, up-milling caused an intermediate impact depth of 400 μm followed by down milling with an impact depth of 300 μm. With merged envelope profiles, it was shown that the effects from rough ceramic milling are gone after 3 finish milling passes, with a total depth of cut of 150 μm.

**Keywords:** high volumetric milling; material removal rate; machining allowance determination; alloy 718; surface integrity; residual stresses

## 1. Introduction

### 1.1. Background and Objective

The planned material volume to be removed from a blank to create the final shape of a part is commonly referred to as allowance. The allowance is dimensioned to give the different stages in the manufacturing chain a reasonably wide process window to protect the processes and the part from undesirable interference [1–3]. Thus, in general terms, the material that constitutes the allowance, and therefore must be removed, is waste and should be minimized from a productivity perspective.

Most ideally, the full depth of the allowance is removed in a single-step operation. However, for most applications, this is too challenging for various reasons. Limitations such as excessive mechanical and/or thermal loads are the most obvious ones as shown in the review of Ezugwu et al. [4]. More subtle reasons, like the release of internal stresses, and, to some degree, the introduction of new stresses, that will cause the part to warp and distort, bring additional justification to the use of multiple steps during removal of the allowance [5]. In practice, these steps are effectively divided into two types of operations, commonly referred to as roughing and finishing. Roughing focuses on material removal

and productivity, while finishing focuses on the performance of the final part, with aspects such as surface appearance and sub-surface properties [6].

A framework for how to assess a suitable machining allowance was suggested in the context of manufacturing an Alloy 718 gas turbine component, using milling in prior works by the authors [7,8]. In those works, the machining effects of the individual milling operations were investigated and a large impact from the rough ceramic milling was found as well as effects caused by tool wear, and thus, change of the cutting edge radii. The machining impact depth was shown to be deeper if residual stresses were considered compared to considering the only deformation. It was also shown that the cutting-edge geometry greatly affected the induced stresses which was believed to be related to increased friction and temperature in the cutting zone using the worn tools. However, in the prior works, the impact from each milling operation was considered separately. In the present work, the combined effect of rough and subsequent finish milling operations is examined.

The objective of the present work is to use a case study to investigate how a prior rough milling operation affects the finish machined surface and, after that, to use this knowledge to design a methodology for how to assess the machining allowance for subsequent milling operations based on residual stresses. Thus, the focus is on the development of methodology rather than drawing general conclusions regarding the milling operations.

### 1.2. Surface Integrity in Relation to Machining Allowance

A component's machining allowance is a well-known industrial concept. It has the intention of leaving a safety margin to the workpiece to ensure that errors and defects from previous steps are removed in later steps. However, the concept of allowance is not well defined in the existing literature. In today's production, machining allowance often only addresses requirements of surface roughness and deformation, measured by hardness changes. Other functional aspects, such as residual stresses, are rarely considered [1]. The machining allowance has a great influence on the process quality and on the efficiency of the manufacturing process as such. If an excessive machining allowance is set, the machining time will increase, which reduce productivity as well as increase consumption of raw materials, tools and energy, thus, increasing the overall processing cost. On the other hand, if the machining allowance is too small, defects and errors from previous process steps will not be removed. In the worst-case scenario, this would jeopardize the structural integrity of the final part. In principle, if the process efficiency is to be improved, the margin should be set as small as possible for the rough machining steps, under the premise that the subsequent finishing steps may ensure correct integrity of the sub-surface of the final part.

Manufacturing often involves several different machining operations where each operation imprints its signature onto the surface. One way of studying the influence of different process steps is to perform subsequent machining operations and leaving a surface area from each step to be analysed. This was done by Dumas et al., who studied the interaction between different turning steps for PH15-5PH steel [9]. The rough turning and two steps of finishing operations showed that no interaction exists if the depth of cut is larger than the previously affected depth. The finishing operations define the final residual stress state. A similar approach was used by Outeiro et al. for orthogonal turning in AISI 304 stainless steel, where the stresses at each of the three passes were evaluated [10]. It was shown that the tensile stresses gradually increased for each pass. However, this result is contradictory to numerical results from FEM simulations from Liu and Guo, whose results showed that tensile stresses from the first cut could alter to compressive stresses for the second cut [11]. The explanation for this was that the affected region of the second layer was thinner due to the change of shear angle and chip thickness, and consequently friction. Similar results were shown by Sasahara et al. who studied the cutting sequence from rough to finish turning of brass by FEM [12]. The results indicated tensile stresses after rough turning, depth of cut of 0.25 mm, which changed into compressive stresses

after finish turning, depth of cut of 0.1 mm. This implies that the negative effects from roughing diminish after one or two finish operations.

The machining allowance for rough machining is especially critical when machining thin-walled parts, which is well covered in the literature [13–15]. D'Alvise et al. used modelling to show the distortion due to the relaxation of thin-walled aeronautical parts [13]. Gao et al. studied milling of thin-wall components and compared FE-modelling with experimental measurements of the geometrical distortion in relation to the milled depth. Liu et al. investigated the influence of depth of cut [15]. The redistribution of residual stresses was investigated for thin wall milling of Aluminium grade 2024-T3 by performing slot milling using different depths of cut. The results showed that as the depth of cut becomes smaller, going from rough to finish milling, the surface residual stresses decrease accordingly. However, the impact from the different subsequent steps was not further investigated or considered. Instead, it was acknowledged that prior residual stress state, e.g., stresses from heat treatment or prior machining, have a great impact on the redistribution of the stresses and that this could be used to tailor the machining of the subsequent steps.

The influence of processing parameters such as cutting speed, feed and depth of cut were shown to have a strong impact when using ceramic and cemented carbide tools, as shown by Akhtar et al. [16]. Similarly, Cai et al. reported on gradually increasing tensile stresses with cutting speed and feed when machining using a solid end mill [17]. The corresponding impact depths increased from 50 μm for low cutting speed (20–25 mm/min) to 100 μm for higher cutting speeds (80–110 mm/min). The resulting strain hardening was influenced by the cutting speed but not the deformation depth. These results showed that the milling operations induced high magnitudes of stress in the cutting zone due to the high temperatures, especially when using dry ceramic milling. To suppress these, various studies have shown promising results using different cooling strategies such as high-pressure cooling, as shown by Polvosa [18], or by using MQL or cryogenic cooling as shown by He et al. [19], Kaynak et al. [20], Shokrani et al. [21] and Pusavec et al. [22]. However, its effects on the resulting integrity of the part is not completely known. Additionally, the resulting surface integrity plays a key role in the fatigue performance of the finished milled surface, as shown by Suárez et al. [23,24]. The results show that compressive stresses and smooth surface result in better fatigue performance. Hence, it is of importance to develop a robust method for determining the machining operations' impact including all processing steps, i.e., from roughing to finishing.

There are some important aspects that need to be considered regarding surface integrity after milling of Alloy 718. Firstly, the surface integrity will vary across the milled path in relation to tool motion, i.e., up, centre or down milling. Selection of milling strategy might be crucial since the deformation and residual stresses can vary greatly. On this topic, the author has shown in prior work that the preferable milling strategy changed with tool wear [8]. A new tool shows lower impact for up milling but as the tool wear increases, down milling shows less deformation and lower tensile stresses. This is due to the redistribution of cutting forces as the tool edge geometry changes. In relation to tool motion and workpiece geometry, Aspinwall et al. observed a difference in surface integrity depending on the milling motion moving upward or downward of an inclined profile [25]. In that study, a solid ball-nosed end-mill was used and the result showed that horizontal upward milling induced compressive surface residual stress (measured parallel to the feed direction), while tensile stress was observed for horizontal downward milling. The reason for the difference was believed to be due to the lower cutting speed for the up milling resulting in a lower localized cutting temperature in the cutting zone. The depth of the compressive residual stress was lower for upwards milling orientation compared to cutting with a horizontal downwards or 0° workpiece tilt angle operation.

Secondly, as shown by Alauddin et al., the tool engagement has a strong influence on the tool life as well [26]. It was shown that full immersion end-milling, compared to semi immersion, resulted in prolonged tool life which also was shown to be greatly affected by the cutting speed. Furthermore, it was shown that down-milling resulted in longer tool

life compared to up-milling. This agrees with the results shown by the author in prior investigations; the increased deformation and tensile stress state alters in the workpiece material depending on up or down-ward milling [8]. On the topic of tool wear, Hadi et al. showed that down milling, on the contrary, resulted in less wear when milling using a tungsten carbide end mill for machining of alloy 718 [27]. Further suppression of tool wear and cutting forces can be achieved with trochoidal milling as shown by Pleta et al., but the induced surface integrity needs to be investigated further [28].

Thirdly, the cutting tool geometry and machining settings play an important role in defining the resulting surface integrity. On this topic, Arunachalam et al. studied the influence of tool edge geometry for turning of hardened Inconel 718 using coated carbide tools with different nose radii [29]. A small edge radius of 0.8 mm resulted in compressive stress while a larger nose radius of 1.2 mm instead induced tensile stress in the surface. This phenomenon is related to friction, and as shown by Grzesik et al., the friction coefficient is significantly higher for large tool nose radii [30,31].

A fourth aspect to consider is the rigidity of the cutting operation which may play a crucial role in the resulting surface integrity. Wojciechowski et al. investigated the interaction between end mill geometry and the resulting surface topography [32]. It was shown that the misalignment from the deflection of the end mill greatly affects the surface topography when machining a hot work tool steel of grade 55NiCrMoV6. The reason was due to redistribution of the cutting forces that may introduce chatter and a roughening of the surface. This is especially critical when long and slender milling tools are used as the deflection becomes larger. The work points out the importance of rigidity in the cutting tool for machining systems to ensure robust machining.

Cutting tool engagement is also important regarding milling kinematics in terms of cutting forces, which was shown by Pimenov et al. who performed face milling tests of a sloping plane of SAE 1045 steel [33]. The machining kinematics changed depending on the relative position of the face mill resulting in a change of cutting force for up- and down milling respectively. This was due to the change of radial immersion of the cutting tool which will lead to redistribution of the cutting force as more of the cutting inserts becomes engaged. This will consequently result in a variation in maximum force per consecutive tooth that may introduce vibrations and increased surface roughness.

In summary, the literature states that it is important to understand the resulting integrity of the surface to optimize the process window, and a common methodology for how to assess and compile the different aspects is still not fully defined. More specifically, there is a need for detailed knowledge of the interaction between processes and workpiece material as well as their combined effect on part performance when applied sequentially as shown by Ezugwu et al. [4,34].

## 2. Materials and Methods

### 2.1. Material

All test samples were taken from a triple melted Alloy 718 casting that was forged in an open die. The forged material was then electrical discharge machined into a rectangular bar. The heat treatment was performed according to AMS 5662, solution annealing at 975 °C for 60 min followed by quenching in a polymer bath. Ageing was done in two temperature steps followed by air cooling. The resulting hardness of the shaft was 261 HB after forging and 432 HB after the final heat treatment. The test piece material had an average grain size of 10–20 μm, see Figure 1, and a hardness of 320 HV1. The alloy composition according to the material certificate is shown in Table 1.

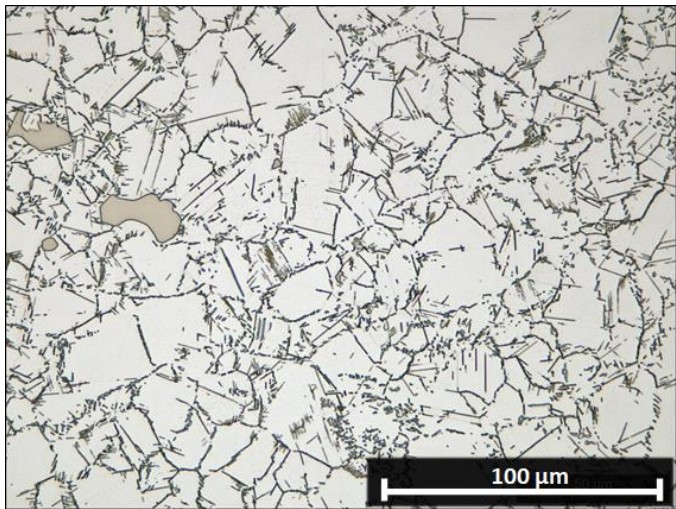

**Figure 1.** Micrograph of the alloy 718 test piece material.

**Table 1.** Chemical composition from material certificate of the alloy 718 billet, in weight-%, used for test specimens.

| Ni | Cr | Fe | Nb | Mo | Ti | Al | Co | C | Mn | P | Si | W | Cu | B | V |
|---|---|---|---|---|---|---|---|---|---|---|---|---|---|---|---|
| 53.9 | 18.0 | Bal. | 5.2 | 2.9 | 1.0 | 0.5 | 0.2 | 0.02 | 0.7 | 0.01 | 0.7 | 0.02 | 0.04 | 0.004 | 0.03 |

*2.2. Methods*

All tests were performed in a Quaser, MV204 II, 3-axis milling machine. The different steps were performed across the length of the sample where the first step was performed using ceramic milling. This was done with a new ceramic insert of grade KYSP30 (RPGN090300E, RPG32E) from Kennametal. The cutting insert had a clearance angle of 11°, a rake angle of 0° but a positive tool engagement angle of 2° and an egg rounding of form E with an egg radius of 57.3 ± 0.3 μm. The 25.4 mm diameter insert tool consisted of a Sandvik BT50 A2B20-50 25 100 insert holder and a Kennametal KIPR025RP09CF03 milling body. The ceramic dry milling was done using a single insert in the tool holder.

After roughing, six different semi-finish milling steps were done using the same tool, a new solid cemented carbide end-mill from Extrica (EX3538T16020) in a Sandvik BT50 A2B14-50 32 070 tool holder. The cutting insert geometry was measured using 3D scanning. The tool had a primary clearance angle of 6.7° and a secondary of 16.6°, a rake angle no. 1 of 55.7° and no. 2 of 57.4°, helix angle no. 1 of 34.1° and no. 2 of 36.4°. The measured egg radius was 11.8 ± 0.1 μm and a nose radius 2.34 ± 0.05 mm. This milling was done using flood cooling and a depth of cut, $a_p$, of 50 μm for each step.

Figure 2 shows, an overview of the ceramic insert after the milling test and the new end-mill before the milling tests. The measurement of the cutting-edge radius was done using the confocal fusion technique using a principle described in prior work [8]. The edge radius was measured at three locations and the presented values are the mean values, including the standard deviations, for the measurements. The tool wear for cemented carbide (CC) milling was minimal, which was also concluded from the surface topography measurements. The same CC tool was used for all of the 50 μm passes. All information regarding the cutting tools is summarized in Table 2.

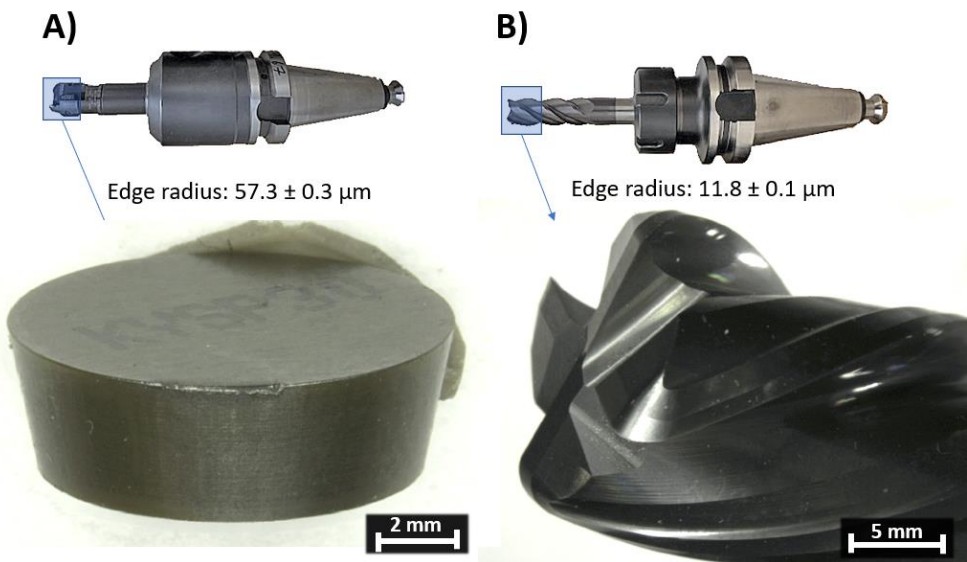

**Figure 2.** Overview of insert and tools used during the tests where the (**A**) Indexable ceramic insert after cutting tests and (**B**) Solid end-mill tool before the cutting tests.

**Table 2.** Details of the cutting tools used for the tests.

| Insert/Tool | Type | Diameter | No. Teeth | Clearance Angles [1] | Rake Angles [1] | Edge Radius [2] |
|---|---|---|---|---|---|---|
| **Material** | | **(mm)** | | **(°)** | **(°)** | **(μm)** |
| Ceramic | Indexable insert | 9 | 1 | 11 | 0 | 56.9 ± 0.2 |
| Cemented carbide | Solid end-mill | 10 | 4 | 6.7, 16.6 | 55.7, 57.4 | 11.8 ± 0.1 |

[1] Measured using 3D scanning of the cutting insert and tool. [2] Measured using confocal fusion method at three locations of the engaged new cutting edge [8]. Reproduced from *Jonas Holmberg, Anders Wretlan, Johan Berglund and Tomas Beno. Selection of milling strategy based on surface integrity investigations of highly deformed Alloy 718 after ceramic and cemented carbide milling. Journal of Manufacturing Processes 2020 58, 193–207.* 

The milling tests were performed across the length of the specimen but stopped to leave a 13 mm long surface, i.e., step, for investigations. Figure 3A shows the resulting milled slot and the different milled steps. Figure 3B shows a schematic illustration of the manufactured staircase samples composed of the seven different steps.

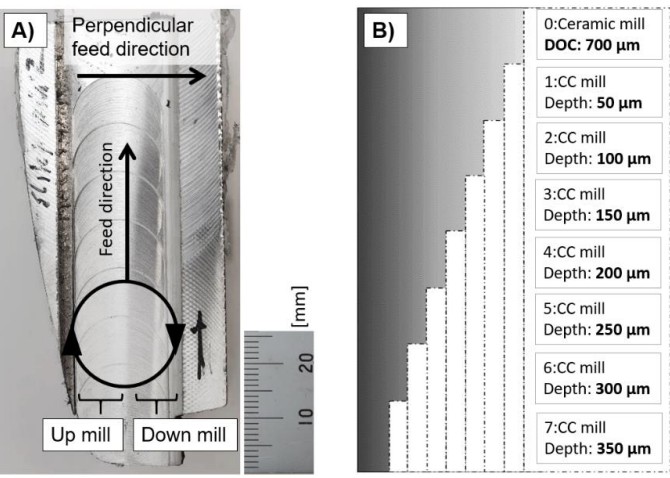

**Figure 3.** (**A**) Overview of the subsequently milled sample including the definition of the milling and orientation. (**B**) Schematic side view of the different milling steps performed on the sample, CC-Cemented Carbide.

In Figure 4 the generated slot, including the definition of the tool motion in the material, is specified. This definition is used in the results section relating to the measured stresses relative to the tool motion. Machining parameters are presented in Table 3.

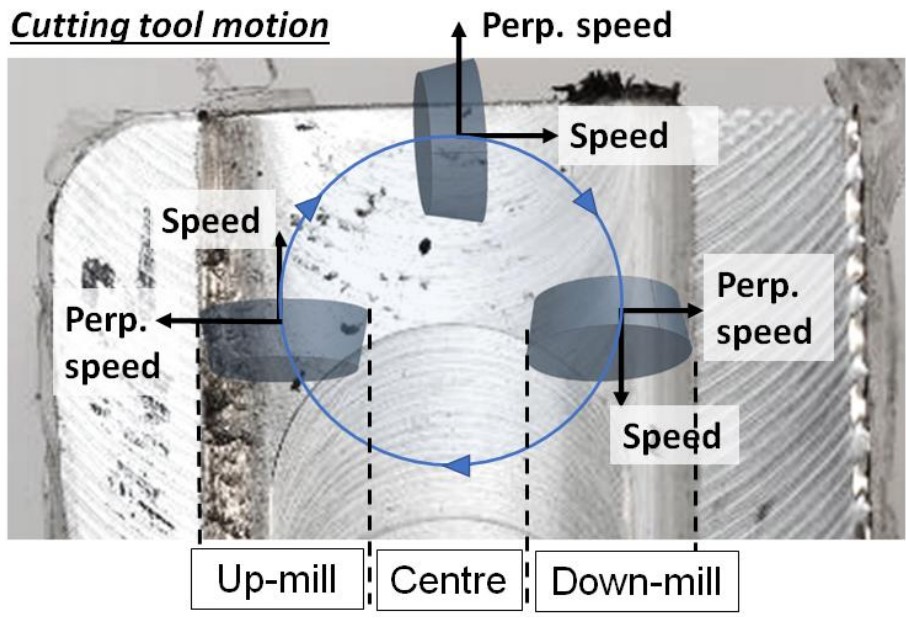

**Figure 4.** Definition of the cutting-edge motion in the material defining the direction of the up/centre/down mill part of the slot.

**Table 3.** Machining settings used for the two types of machining operations.

| Insert/Tool | $v_c$ | $v_c$, Engage [1] | $v_f$ | $s$ | $f_z$ | $a_p$ |
|---|---|---|---|---|---|---|
| **Material** | **(m/min)** | **(m/min)** | **(mm/min)** | **(rev/min)** | **(mm/tooth)** | **(mm)** |
| Ceramic | 800 | 500 | 611 | 10,000 | 0.06 | 0.7 |
| Cemented carbide | 50 | 50 | 127 | 1590 | 0.02 | 0.05 |

[1] $v_c$ at engaged diameter.

### 2.3. Evaluation Methods

The main objective of the surface topography analysis was to study if tool wear had any significant influence on the machined surface texture. The surface topography after milling was measured with coherence scanning interferometry using a Sensofar S Neox instrument at $10\times$ magnification. Measurements were performed over a $5.7 \times 4.29$ mm stitched area, with an overlap of 15%, in the centre of the sample at one position with a lateral, x-y plane, uniform resolution of 1.29 µm. The measured data were analysed with the software MountainsMap from Digital Surf. Form removal was done with a 2nd order polynomial and the data were filtered using a spatial median noise-reduction filter to reduce short-wavelength noise, window size $5 \times 5$ points. An L-filter was not used, thus, S-F surfaces were analysed.

The topography was evaluated according to the ISO 25178-2:2012 standard [35]. Selected parameters were used to give a reasonable description of the surface texture. The selected and calculated parameters were: the arithmetic mean height, Sa, foremost describing the longer wavelengths of the surface roughness since these usually also have the highest altitude, the developed interfacial area ratio, Sdr, to give additional information of the roughness regarding the short wavelengths of the texture, and Ssk, to study the skewness of the height distribution [35].

Residual stresses were measured by X-ray diffraction, using an XStress 3000 G2R diffractometer from Stresstech equipped with an Mn X-ray tube ($\lambda$ = 0.21031 nm). X-ray diffraction measures the interplanar spacing in the atomic lattice. The modified $\sin^2\chi$ method was used with $\pm 5$ tilt (psi) angles ($45°\ldots-45°$) and the (311) lattice plane located at 151.88° was evaluated. Residual stress was calculated assuming elastic strain theory according to Hooke's law using 199.9 MPa as Young's modulus and 0.29 for Poisson's ratio, as described by Noyan and Cohen [36].

Layer removal was done to obtain residual stress profiles. This was performed with successive material removal by electro-polishing with a Struers Movipol and A2 electrolyte. All measurements were performed in an accredited laboratory in accordance with the EN 15305:2008 standard [37]. The evaluation of the measured data is connected to an error related to the linear regression of the measured diffraction peak at different tilting angles, defined by the modified $\sin^2\chi$ method. This error has been presented for the surface measurement as error bars in the diagrams. However, for the stress profiles, this error is not added to the result since the error is low in relation to the stresses, typically less than 20 MPa, and will not be visible in those figures. Other errors to take into considerations are connected to the method itself, this includes the operator and material to be measured. These errors are in the interval 10–20 MPa, where the main source of variation comes from the material to be measured as shown in prior work [38].

The diffraction peaks were evaluated using the methodology described by Prevéy by peak fitting of the K$\alpha$1 and K$\alpha$2 peaks [39]. This was done using a parabolic background a Pearson VII for peak fitting.

### 3. Results

*3.1. Surface Topography*

The main aim of the surface typography measurements was to examine that no significant tool wear occurred during the machining of the seven different cemented carbide passes. The results show only a marginal difference between the different cemented carbide steps, seen in the 3D representation of the milled surfaces in Figure 5. The corresponding topography parameters *Sa*, *Sdr* and *Ssk*, presented in Table 4, indicate that both *Sa* and *Sdr* have similar values for cemented carbide milling and significantly lower values for ceramic milling. The *Sdr* value, which describes the complexity of the surface is significantly lower for ceramic milling, indicating a dissimilar texture. It is further observed that *Ssk* changes from negative values for the first steps to positive for the last two steps. This indicates that the height distribution gradually changes from above the mean plane (peaks) to below the mean plane (valleys) for the 300 and 350 µm steps.

**Table 4.** Topography results for the ceramic and cemented carbide milled steps after 50–350 µm milling.

| Milling Operation | Step (µm) | *Sa* (µm) | *Ssk* (-) | *Sdr* (%) |
|---|---|---|---|---|
| Ceramic | 0 | 0.36 | 0.03 | 0.06 |
| Cemented carbide | 50 | 0.62 | −0.33 | 0.80 |
| Cemented carbide | 100 | 0.46 | −0.04 | 0.59 |
| Cemented carbide | 150 | 0.63 | −0.23 | 0.76 |
| Cemented carbide | 200 | 0.64 | −0.21 | 0.84 |
| Cemented carbide | 250 | 0.56 | −0.10 | 0.69 |
| Cemented carbide | 300 | 0.51 | 0.35 | 0.64 |
| Cemented carbide | 350 | 0.56 | 0.34 | 0.71 |

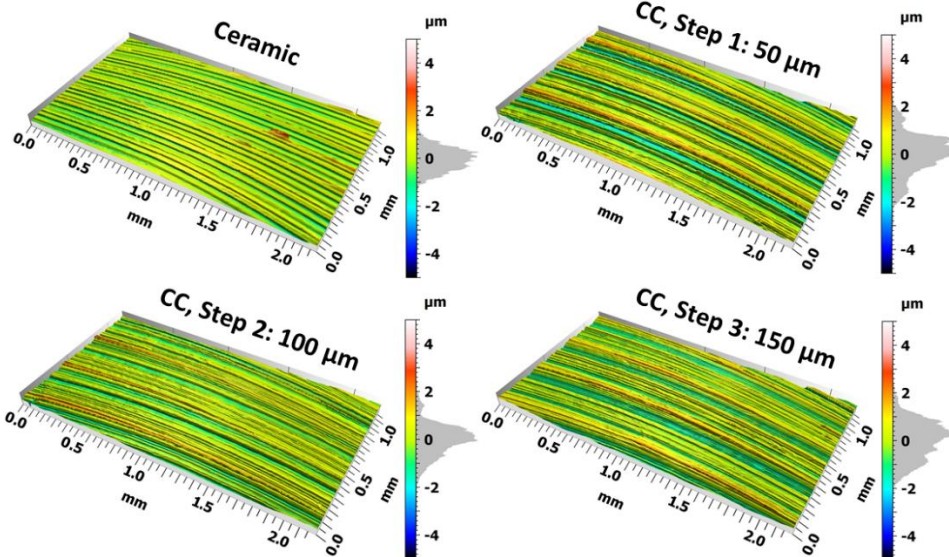

**Figure 5.** 3D topography illustrations of the first four of the seven milled steps, after ceramic milling (CC), cemented carbide milling after 50 μm, 100 μm and 150 μm.

### 3.2. Residual Stress

The surface residual stresses differ between the different milling steps. In Figure 6 the surface stresses for the up-, centre and down milling part of the slot for the different steps are shown.

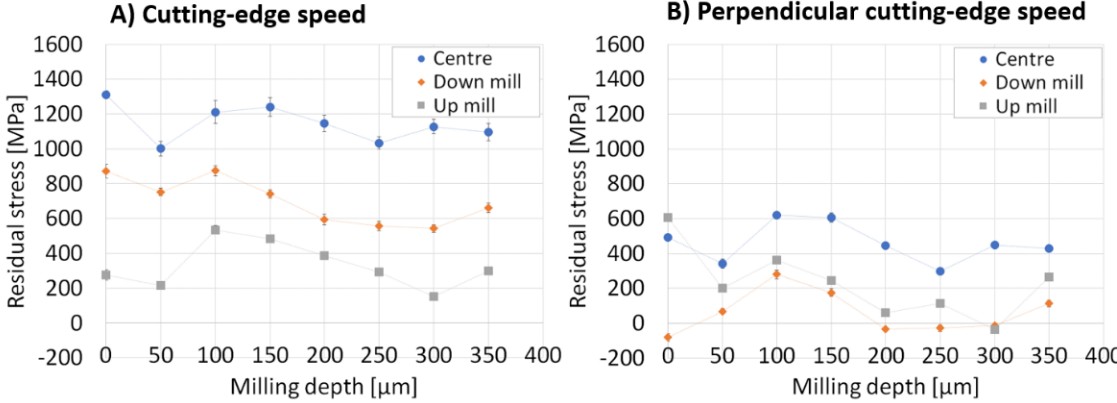

**Figure 6.** Surface residual stresses at up-, centre and down-milled part of the slot. The error bars represent the errors from linear curve fitting of diffraction data.

The results indicate that the centre of the slot for ceramic milling experiences the highest tensile stresses for the cutting-edge speed direction. It is also observed that cemented carbide milling has induced just as high, or in some cases higher, tensile stresses as ceramic milling. A decay of tensile stresses in the surface with the different steps was also observed.

In the other direction, perpendicular to cutting edge speed, the stresses are comparably lower and there is a slight tendency of decaying tensile stresses with the different steps. The centre of the slot shows the highest values but both up- and down milling parts have similar stresses for the deeper steps, towards the right in the diagrams. Closer to the surface, towards the left in the diagrams, the lowest tensile stresses are seen for ceramic down milling. The first cemented carbide (CC) milled step results in significantly lower tensile stresses for the centre and up milled part.

The residual stress profiles for ceramic milling at up, centre and down milled parts of the slot are shown in Figure 7. Generally, for all three positions, the profile is similar

and characterized by high tensile stresses in the surface that drastically drop below the surface. The centre and down milling part show similar trends for the surface where the cutting-edge speed direction has the highest tensile stresses while up milling has the highest tensile stresses in the surface for the perpendicular direction. Below the surface, there are higher tensile stresses for the cutting-edge speed direction.

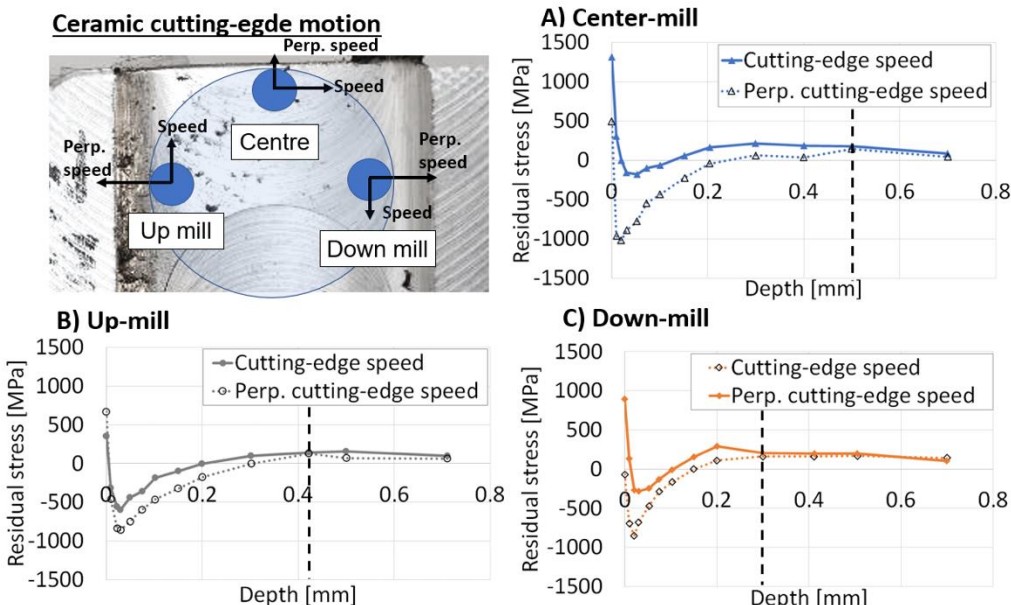

**Figure 7.** Residual stress profiles from surfaces machined with new ceramic insert at different positions across the slot.

It is further shown that the up-, centre and down-milling has resulted in different stresses and impact depths. The centre milling has induced the highest surface stresses and the deepest impact. The up-milling shows the lowest impact and down milling is intermediate. It is further observed a distinct difference where the directional influence diminishes, marked with dashed lines, which is observed at a depth of 500 μm for centre milling and 400 μm and 300 μm for up- and down milling respectively. It is also seen that at these depths a steady state is reached.

In Figure 8, the residual stress profiles after the first 50 μm cemented carbide milled step is shown. Generally, there is a greater difference in the profile characteristics for these positions compared to the ceramic milled slot. Again, centre milling has resulted in the most severe impact with high tensile stress for the surface followed by a more gradual decrease, compared to ceramic milling, into the compressive zone. Similar to ceramic milling, up-milling shows the lowest stresses but the deepest impact. Generally, the depth where the directional impact diminishes is much smaller for all positions compared to ceramic milling. The centre and down milling have a depth impact of less than 100 μm while up-milling is difficult to assess since a 50 MPa offset is seen throughout all depths.

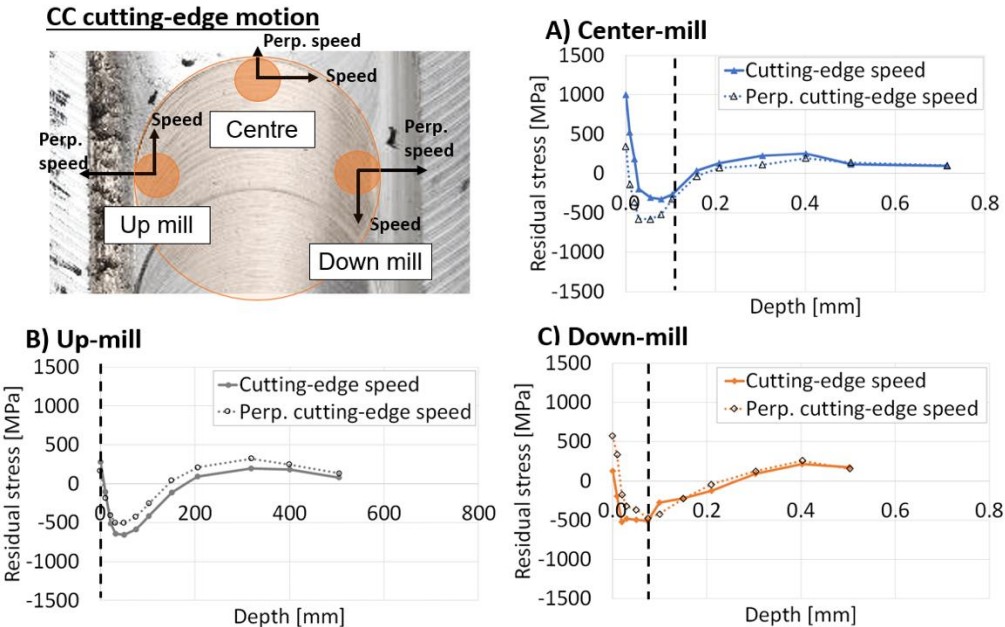

**Figure 8.** Residual stress profiles from surfaces subsequently machined first with ceramic milling followed by 50 μm step with a new cemented carbide tool (CC) at different positions across the slot.

Regarding the total affected depth, where the core state is reached, it is observed to be far greater than the depth of diminishing directional influence. For all three positions, it is observed that the stress balancing tensile stress peak gradually levels out at a depth of 500 μm.

Additionally, residual stress profiling for all seven steps is shown in Figure 9. In this figure, the different profiles have been separated by the milled depth to create an offset in relation to the final geometry. These figures show the difference clearer on the surface as well as how the maximal compressive stress differs between the different steps. Generally, all profiles have similar characteristics with high tensile stress in the surface region which drastically drops into compressive stresses below the surfaces. The maximum compressive stress is typically reached at a depth of 80–120 μm. For greater depths, the compressive stresses decrease and changes into tensile stresses and the core state of 50–200 MPa is reached for depth below 300 μm. The profiles do however differ in terms of magnitude and location of zones of transitions from tensile to compressive stress and stress maximums.

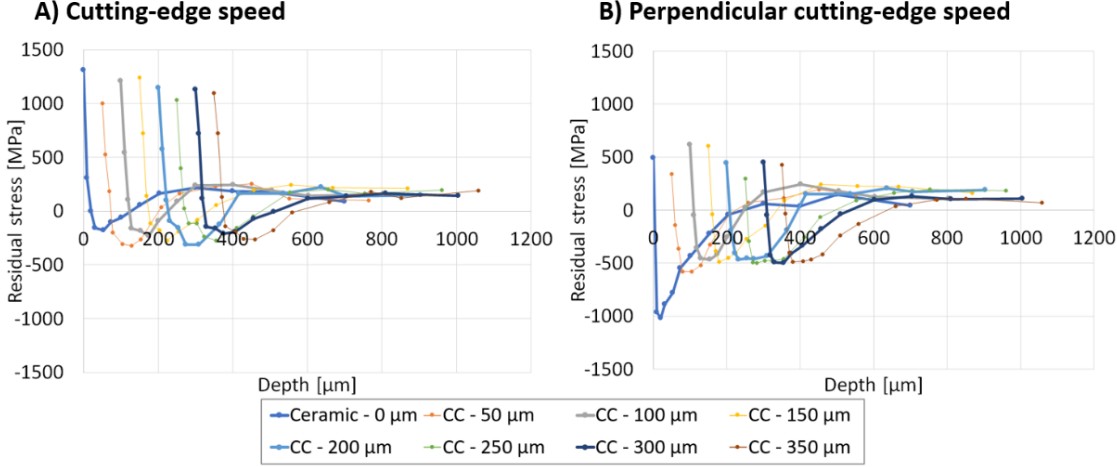

**Figure 9.** Residual stresses for each milling steps for the centre of the slot.

For the cutting-edge speed direction, the ceramic milled profile is very similar to the cemented carbide milled profiles. However, the high tensile stresses are counterbalanced by the lowest compressive stress profile "valley", followed by a low and wide tensile stress peak located at 200–300 µm. For the perpendicular direction, the ceramic milled profile is clearly distinguished by the deepest compressive stress profile "valley" which is followed by a slow change into the core state. In this direction, no tensile stress balancing peak is observed for the ceramic milled profile.

The cemented carbide profiles are very similar but can be grouped in two main categories based on the shape of the profiles: the profiles for milling steps of 50–150 µm have a sharper transition into tensile stress peak at depth of 100 µm compared to the 200–350 µm profiles. The characteristics of the 200–350 µm profiles instead show a shift with deeper impact and a more gradual decrease of compressive stresses into the core stress state. It is also observed that the tensile stress balancing peak is becoming lower with the different cemented carbide milling steps.

Merged envelope profiles have been created by accumulating the highest tensile and compressive stress values for all seven steps for the centre of the milled slot. These envelope profiles are shown in Figure 10. Illustrating the results in this way makes it easier to determine when a steady state is attained, which occurs earlier for the speed direction compared to the perpendicular direction. It is shown that the deep compressive stress balancing effects from ceramic milling levels out at a depth of 100–150 µm below the surface.

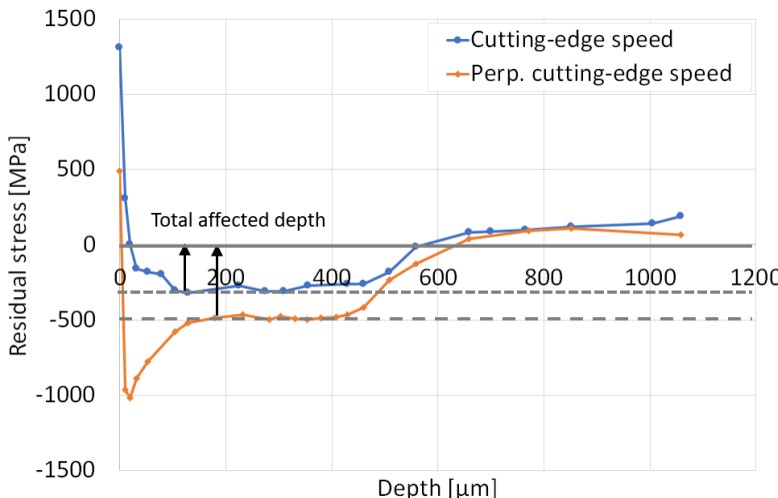

**Figure 10.** Envelope profiles, representing the accumulated profiles for all the maximum (compressive and tensile) values for all steps at the centre of the slot.

## 4. Discussion and Analysis of Results

The investigated milling operations in this work were rough milling with a ceramic insert and finish milling with a solidly cemented carbide end mill. The finish milling is considered to generate the final functional surface and the selection of the small $a_p$ was done based on the ability to catch the transition of when the effects of ceramic rough milling diminishes. However, the relatively small $a_p$ will result in a small tool engagement. In principle, only the cutting-edge will be engaged during the cutting. This might not be preferable for the cutting tool in terms of a chip breakage point of view, but still, a relevant depth of cut when considering finish milling.

The residual stress profiles show a clear influence from the respective milling steps. However, all profiles have similar characteristics where the ceramic milling, as expected, induced the highest and deepest magnitudes. The magnitudes are in line with others research and explained with the high thermal load that is developed in the cutting zone [16,17].

Regarding rough milling strategy, ceramic up-milling results in the lowest tensile surface stresses, while the depth impact is lower for down milling and in this perspective, down milling, is preferable. The corresponding results for cemented carbide finish milling show instead both the lowest surface stresses and depth impact for up-milling. This is in line with the results shown by Pimenov et al. who showed that up-milling results in lower cutting forces and less vibration attributed to the smoother engagement of the cutting tool as it gradually is immersed into the workpiece [33]. Generally, cemented carbide milling has much lower impact depths compared to ceramic milling. These results are related to the tool engagement and accessibility of the cooling for the case of finish milling. This, in combination with the chip's formation, will allow for accessibility of the coolant during the machining. The influence of other cooling media on the resulting stress profile would be an interesting continuation of this work where high pressurized cooling or cryogenic cooling would possibly lower the temperatures in the cutting zone and thereby reducing the tensile stresses [18].

Analysing the results further for the development of an allowance determination strategy, it is suggested that the impact depth could either be determined as the depth position where the core stress state is reached or the depth where the directional influence from the milling diminishes, i.e., the depth where the stresses for the two measurement directions coincide. As suggested in this work, using the directional diminishing depth and merging the different steps together, makes it possible to assess how deep the superimposed effects from the different milling steps are. In Figure 10, the accumulated envelope profiles for all steps show that a steady state occurs after 100 µm cemented carbide milling and that the effects of the tensile stress balancing region occur after 150 µm cemented carbide milling. This is visible as a plateau in Figure 10, starting at depth of 100 µm in speed direction and at 150 µm in perpendicular to speed.

In Figure 11, the merged profiles of the ceramic rough milling and the first step of finish milling have been constructed for the three different positions across the slot. For comparison, the ceramic milling profiles were also added in these figures, seen as dashed profiles. The results show that the directional influence diminishes much closer to the surface compared to ceramic milling. It is observed that centre milling has a directional diminishing depth impact of 150 µm and down-milling 200 µm. The up-milling does not show any directional difference in this manner but instead shows a 100 MPa profile offset for all depths below 10 µm. Summarizing the results, it becomes obvious that up-milling is the preferable milling strategy for the selected cutting tools. It is also shown that by using this approach, it is possible to study the influence of different subsequent operations in the same profile.

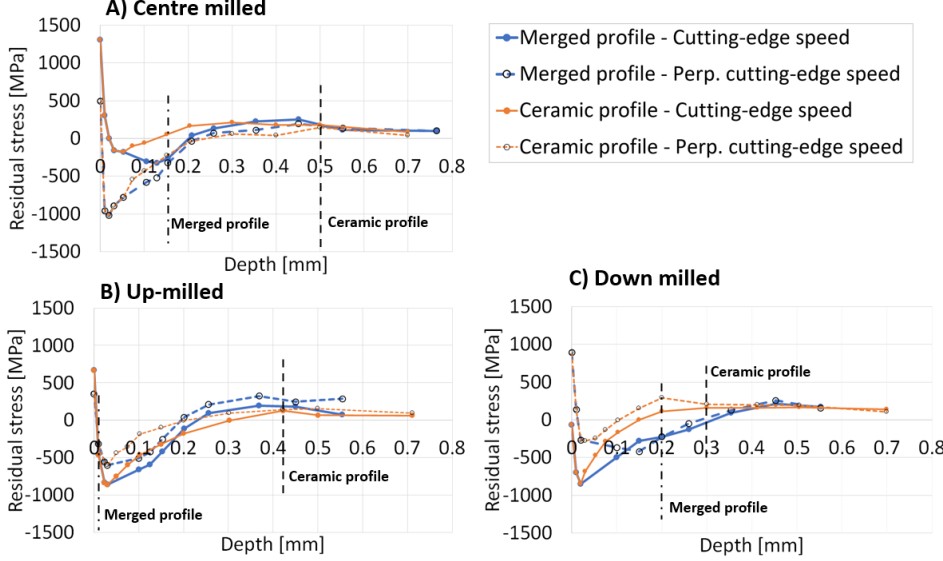

**Figure 11.** Ceramic and evolution (envelope) profiles, ceramic and 50 µm CC milling at different locations across the slot.

In relation to deformation, it has been shown in prior work that deformed grain boundaries could be observed to depths of 40 µm for ceramic milling and 6 µm for cemented carbide milling [8]. This is much less than the assessed impact depth determined based on the residual stresses shown in this work. This further strengthens the argument to use residual stress profiling when assessing the machining allowance.

One research question that initiated this work was to study how deep the rough milling effects are. The merged envelope stress profiles show that the rough milling effects diminish after 2–3 finish milling steps using an $a_p$ of 50 µm. This is in parity with Sasahara et al. who showed in FE simulation that the effects of turning diminished after 1–2 finish cuts equal to a depth of 0.2 mm [12]. However, after removing the rough milling effects, the finish milling operation itself has induced high surface tensile stresses and significant impact depths. This needs to be considered if the surface is to be used as a final and functional surface and measures might need to be taken to suppress the tensile stresses. Such measures could be a selection of machining parameters, coolant, coolant supply or selection of milling strategy. In this work, it has been shown that up-milling is preferable for cemented carbide milling which is in line with prior investigations [8]. However, as the prior study has shown, the tool wear might alter this and shift into lower impact for down-milling instead. This is was also concluded regarding tool wear by Hadi et al. [27] but the chip formation was noticeably different for up- and down milling.

The other research question in this work was to evaluate a methodology for how to assess the machining allowance based on residual stresses that consider prior process steps. This methodology assesses the machining allowance considering two criteria: (1) the depth where the core residual stress state is reached, (2) the depth where the effects of directional difference diminish.

Based on the work in the present paper, to assess the machining allowance, it is suggested to perform milling tests and evaluate the residual stresses of a stepwise milled slot. Using this approach, it is suggested to make decisions based on the residual stress depth impact from different operations. It was shown that stresses along feed and perpendicular to feed change with depth and that the difference between the directions diminishes at a specific depth. The directional diminishing is related to the material strength and when the material fulfils the plastic flow criteria. As shown in prior work, the ceramic milling process results in a severely deformed surface with nano-crystallization and a greater depth impact and a clear plastic material flow can be seen [8]. This material flow is in the feed direction, i.e., the direction of the machining tool path, and will diminish with depth. This deformation will consequently induce a residual stress signature, which is also defined by the direction of the material and tool path. This will cause an redistribution of stresses and result in a stress balancing gradient in three dimensions. In the present work, it has been shown that the stresses across and along the tool path will be greatly different because of this plastic flow. However, this effect will diminish with depth and as the two stress directions coincide, a limit of plastic flow is attained. This limit relates to the depth to which the heat impact has affected the workpiece material. For ceramic milling, this depth is similar to the total affected depth while it is not for the cemented carbide milling. To further separate the difference between the different strategies, it is suggested to evaluate the heat impact further. This could be realised by means of embedded thermocouples in the workpiece during machining. Such investigation would clarify the superimposed effects from the rough milling difference in terms of material flow and deformation and it is suggested to do an in-depth investigation of the deformation.

The knowledge in this work is also possible to transfer to other milling operations such as planar milling and could be especially useful to determine the effects for more advanced milling strategies such as trochoidal milling [28]. In that case, one needs to design the milling tracks in an optimal way and, as shown in this work, should be done by selecting a milling path to sidestep by leaving the up-milled part as a finished surface. However, as shown by prior research this might alter as the cutting tool wear increase [8].

Further evaluation of different machining processes is needed to establish a robust methodology for assessing machining allowance based on the diminishing plastic flow criterion. It is suggested to perform mechanical testing to investigate the relationship between mechanical strength and plastic material flow from the milling processes. This is also related to the fatigue properties. Depending on the application of the finished surface, the fatigue properties might be crucial. As shown by Suárez et al., the machining strategy may play a key role in determining the fatigue life of the finished surface [10,11]. One natural final step in establishing this allowance methodology would be to test it in relation to fatigue. Additionally, for future work, the influence of both ceramic and CC tool wear should be included since that has a strong impact on the resulting residual stresses as well.

The results in this investigation could be used as guidance when defining the manufacturing route and how to assess the impact from different subsequent operations. The intention has been to show how prior process steps affect later process steps. The methodology and results could both be used to determine appropriate milling strategy regarding the tool motion, up- or down-milling, but also when selecting the machining allowance for rough machining.

## 5. Conclusions

This investigation has considered the influence of rough ceramic milling on the subsequent milling operations and the following conclusions have been made:

- Both rough ceramic milling and finish cemented carbide milling induce high tensile stresses that are counterbalanced by a high and deep compressive stress zone below the surface followed by a tensile stress peak.
- The lowest surface residual stress impact for ceramic rough milling was found for up-milling but down milling resulted in lower impact depth. Finish up-milling resulted in the lowest surface stress and residual stress depth impact.
- The depth impact after rough ceramic milling diminished after 2–3 finish milling steps equal to a total depth of cut of 150 μm.
- Measurements of the depth of diminishing of directional influence could be used to assess the impact depth from prior millings steps which correlated well with the total impact depth for ceramic milling but not for cemented carbide milling.
- Based on the diminishing of directional influence impact after ceramic rough milling, the least impact depth was shown for down-milling, 300 μm, and the deepest impact was shown for centre milling, 500 μm. The corresponding influence for cemented carbide milling showed no directional difference at up-milling while centre and down milling resulted in shallow impact depth of 100 μm.

**Author Contributions:** Conceptualization, J.H., A.W., A.M.K. and J.B.; methodology, J.H.; validation, J.H., formal analysis, J.H. and A.W.; investigation, J.H.; resources, A.W. and A.M.K.; data curation J.H., A.W., and A.M.K.; writing—original draft preparation, J.H.; writing—review and editing, J.H., A.W., J.B., T.B.; visualization, J.H.; supervision, A.W., J.B. and T.B.; project administration, J.H.; funding acquisition, J.H. All authors have read and agreed to the published version of the manuscript.

**Funding:** This research was funded by VINNOVA, Sweden's innovation agency, with grant number [2018-02976].

**Institutional Review Board Statement:** Not applicable.

**Informed Consent Statement:** Not applicable.

**Data Availability Statement:** The data presented in this study are available on request from the corresponding author.

**Acknowledgments:** Thanks for the financed by VINNOVA, Sweden's innovation agency. Special thanks to GKN Aerospace Sweden AB and Tooltec Trestad AB. The authors also would like to thank the KK-foundation and the SiCoMaP research school.

**Conflicts of Interest:** The authors declare no conflict of interest.

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
