# Peer review of "Surface Integrity Investigation to Determine Rough Milling Effects for Assessment of Machining Allowance for Subsequent Finish Milling of Alloy 718"

_jmmp, doi:10.3390/jmmp5020048_

Round 1

Reviewer 1 Report

The review concerns the work entitled: „Surface integrity investigation to determine rough milling effects for assessment of machining allowance for subsequent finish milling of alloy 718”.

Main comments are below:

Paragraph 2.1. Material.
Could the Authors add more detail - properties of studied material?

Page 7, line 225. The number of paragraph titled 'Evaluation methods' is wrong. It should be 2.3.

Moreover - What % of surface overlap did the Authors assum for stiching? 
What lens did the Authors use in the topography studies?
On what basis did the Authors selekt the surface texture parameters for the analysis: Sa, Ssk and Sdr?

Paragraph 3. Results.
Title of paragraph 3.1. "Topograpgy" - it should be "Surface topography"
These results need more descriptions (Table 4, Figure 5).
Figures 6-9 and figure 11. Descriptions above the figures are redundant, as the descriptions are included in the figures titles.

Author Response

The review concerns the work entitled: „Surface integrity investigation to determine rough milling effects for assessment of machining allowance for subsequent finish milling of alloy 718”.

Main comments are below:

  1. Paragraph 2.1. Material.
    Could the Authors add more detail - properties of studied material?

Action: Additional information of the grainsize, hardness and micrographic image of the material has been added to the manuscript. In the Table text, a comment regarding the chemical information from certificate has been added as well.

  1. Page 7, line 225. The number of paragraph titled 'Evaluation methods' is wrong. It should be 2.3.

Action: Thanks for the comment, this has been adjusted in the manuscript.

  1. Moreover - What % of surface overlap did the Authors assum for stiching? 
    What lens did the Authors use in the topography studies?
    On what basis did the Authors selekt the surface texture parameters for the analysis: Sa, Ssk and Sdr? – Just to study inf the tool wear had any significant influence on the surface state.

Action: Comments regarding this has been added to the manuscript under experimental and results sections.

  1. Paragraph 3. Results.
    Title of paragraph 3.1. "Topograpgy" - it should be "Surface topography"

Action: This has been adjusted in the manuscript.

  1. These results need more descriptions (Table 4, Figure 5).

Action: This has been adjusted in the manuscript

6. Figures 6-9 and figure 11. Descriptions above the figures are redundant, as the descriptions are included in the figures titles.

Action: The figure text has been adjusted, for clarity I want to keep the text in the images instead.

Reviewer 2 Report

The paper shows an interesting analysis work on the residual stress generation during Inconel 718 milling. Some points should be modified or enhanced to increase the overall quality of the paper for its publication.

English should be reviewed throughout the paper, some grammar errors and unfinished sentences can be found.

Introduction

The bibliography seems out of date. The more classical references are included but some more recent works are missing, especially on a topic as nickel-based alloy machining and surface integrity which generates an extensive number of papers every year.

The section concerning the definitions for ‘Allowance‘ and their clarification feels kind of confusing. It is not clear the reason for this clarification concerning the storyline of the paper or its goals. Please, consider revising it to achieve a higher clarity.

Methods

The grade of the ceramic insert should be specified in the text (KYSP30 from figure 2?)

The commercial reference for the ceramic toolholder should be included.

Geometry for the cutting tools is not clear:

  • In table 2 it is stated 9 mm diameter for the ceramic tool and 10 mm for the carbide tool. The ceramic insert geometry is RPGN090300E: 9.525 mm diameter round positive insert. The diameter for the ceramic tool can be calculated from the data in table 3 (25.4 mm), but it could appear when introducing the toolholder. Please, consider exposing these values explicitly for clarity.
  • Other cutting geometry data is missing as rake and helix angles for both tools or the nose radius for the carbide tool. They should be included too.

Concerning the wear on the carbide tool, no actual wear is appreciated in figure 2, it seems the image of a new tool. Was a new tool employed in each 50 µm depth pass? If not, this tool would be engaged during several minutes to perform all the 50 µm steps and some visible wear would be expected.

Evaluation methods

The lateral resolution in both X and Y directions is the same (1.29 µm) in the SNeox instrument? If it is not the same, both values should be included.

The use of Sa parameter is extended, but the reasons for using skewness (Ssk) and area ratio (Sdr) and not others (for example kurtosis Sku), for characterizing the surfaces should be exposed in the paper.

ISO 25178 part 2 defines different operators to be applied on a surface to obtain S parameters. The 2nd order polynomial would correspond to an F operator and the 5x5 window filter to a S operator. Why was not a L operator applied to obtain the S parameters? Why was a window filter applied to remove the high-frequency noise instead of a wavelength based filter (e.g. gaussian or spline)?

Concerning the experimental errors for XRD measurements, it is stated a 20 MPa error related to the stress estimation based on the sin2χ method. Besides this error, was any measurement preformed more than one time to evaluate the experimental variability for these measurements? This variability should be identified and evaluate its possible impact on the obtained results, especially in the comparison of the stress profiles from consecutive carbide milling passes performed in the results section (Figure 9). These profiles are quite similar and the differences could be bellow the experimental variability for the measurement of one profile.

Results

There is quite a deep analysis and discussion concerning the residual stress results. Concerning the topography results, instead, only the value exposition is performed. This makes the topography results feel like not providing actual value to the discussion and the paper. Can some deeper analysis be performed concerning the topography results? If not, author may consider removing it and focus only on the residual stress analysis.

Discussion

The milling operation tested is a slot milling operation. In a planar milling operation, the tool path would show some overlapping and the cutting process would not have the up or down milling section. Can the results obtained in these tests be extrapolated to a planar milling condition or the obtained results and conclusions should be limited to grooving operations? Some discussion should be provided in this points to clearly identify or limit the applicability of the results from the paper.

If the carbide tool showed no wear, how can the results obtained be related to an actual industrial machining process? Could the average tool wear present in a tool modify the results obtained in the paper concerning the consecutive passes? Maybe make the need or more or less passes to remove the effect from the ceramic milling? Would it make sense a future working line to identify the tool wear range that keeps these results consistent along a time lengthy machining process?

Author Response

Comments and Suggestions for Authors

The paper shows an interesting analysis work on the residual stress generation during Inconel 718 milling. Some points should be modified or enhanced to increase the overall quality of the paper for its publication.

  1. English should be reviewed throughout the paper, some grammar errors and unfinished sentences can be found.

Action: The manuscript has been reviewed externally for grammar and language and adjusted accordingly.

Introduction

  1. The bibliography seems out of date. The more classical references are included but some more recent works are missing, especially on a topic as nickel-based alloy machining and surface integrity which generates an extensive number of papers every year.

Action: Additional more up to date references has been added to the manuscript. 

  1. The section concerning the definitions for ‘Allowance‘ and their clarification feels kind of confusing. It is not clear the reason for this clarification concerning the storyline of the paper or its goals. Please, consider revising it to achieve a higher clarity.

Action: This section of the introduction has been narrowed down for clarity. Machining allowance is one of the main purposes of the manuscript, but I agree that the text could be simplified according to your comment.

Methods

  1. The grade of the ceramic insert should be specified in the text (KYSP30 from figure 2?)

Action: Yes, correct - this has been adjusted in the manuscript

  1. The commercial reference for the ceramic toolholder should be included.

Action: This has been added in the manuscript

  1. Geometry for the cutting tools is not clear:
  • In table 2 it is stated 9 mm diameter for the ceramic tool and 10 mm for the carbide tool. The ceramic insert geometry is RPGN090300E: 9.525 mm diameter round positive insert. The diameter for the ceramic tool can be calculated from the data in table 3 (25.4 mm), but it could appear when introducing the toolholder. Please, consider exposing these values explicitly for clarity.

  • Other cutting geometry data is missing as rake and helix angles for both tools or the nose radius for the carbide tool. They should be included too.

Action: This has been added in the manuscript

  1. Concerning the wear on the carbide tool, no actual wear is appreciated in figure 2, it seems the image of a new tool. Was a new tool employed in each 50 µm depth pass? If not, this tool would be engaged during several minutes to perform all the 50 µm steps and some visible wear would be expected.

Action: The same tool was used. Measurement of the cutting edge was done showing minimal tool wear. The image in Figure 2 is however for the new tool, unfortunately we lost track of the worn tool. I have made this clear in the text and figure text. No effect of tool wear could be seen in the milled surface textures.

‘Evaluation methods

  1. The lateral resolution in both X and Y directions is the same (1.29 µm) in the SNeox instrument? If it is not the same, both values should be included.

Action: Yes, the same resolution in X-Y, this has been clarified in the manuscript text.

  1. The use of Sa parameter is extended, but the reasons for using skewness (Ssk) and area ratio (Sdr) and not others (for example kurtosis Sku), for characterizing the surfaces should be exposed in the paper.

Action: The selected parameters were used to give a simple but reasonably complete description of the surface topography in order to study if any effects from tool wear. This has been clarified in the manuscript.

  1. ISO 25178 part 2 defines different operators to be applied on a surface to obtain S parameters. The 2ndorder polynomial would correspond to an F operator and the 5x5 window filter to a S operator. Why was not a L operator applied to obtain the S parameters?

Action: ISO 25178 does not stipulate that an L-filter must be used. In our case, we thought it would serve our purpose best to analyse S-F surfaces.

  1. Was a window filter applied to remove the high-frequency noise instead of a wavelength based filter (e.g. gaussian or spline)? Filter:

Action: The filter selection was done on basis from prior knowledge that we consider this to be an effective filter to be used in this application.

  1. Concerning the experimental errors for XRD measurements, it is stated a 20 MPa error related to the stress estimation based on the sin2χ method. Besides this error, was any measurement preformed more than one time to evaluate the experimental variability for these measurements? This variability should be identified and evaluate its possible impact on the obtained results, especially in the comparison of the stress profiles from consecutive carbide milling passes performed in the results section (Figure 9). These profiles are quite similar and the differences could be bellow the experimental variability for the measurement of one profile.

Action: This is a correct observation. We have performed through MSA analyses of this system and the results clearly show that the main source of error comes from variation in the sample material itself. For an ideal sample, flat sample of shot peened steel, we manage to get and total system variation of 10 MPa, but in this case the variation is probably higher. In this case we repeated the measurement on the surface and saw low variation however, to state the that in the manuscript would require an extensive analyse and investigation which is almost a manuscript itself. However, I have added a reference regarding the measurement uncertainty.

Results

  1. There is quite a deep analysis and discussion concerning the residual stress results. Concerning the topography results, instead, only the value exposition is performed. This makes the topography results feel like not providing actual value to the discussion and the paper. Can some deeper analysis be performed concerning the topography results? If not, author may consider removing it and focus only on the residual stress analysis.

Action: The objective with the surface topography measurements was to study effects from tool wear. This clarification has been added to the manuscript.

Discussion

  1. The milling operation tested is a slot milling operation. In a planar milling operation, the tool path would show some overlapping and the cutting process would not have the up or down milling section. Can the results obtained in these tests be extrapolated to a planar milling condition or the obtained results and conclusions should be limited to grooving operations? Some discussion should be provided in this points to clearly identify or limit the applicability of the results from the paper.

Action: Thanks for the comment, this is knowledge that could be adapted to optimise the milling track design of planar milling operations. These results show that one should avoid centre milling and instead use the up-milling part of the tool when planning the milling. This comment has been added to the discussion/analysing part of the manuscript.

  1. If the carbide tool showed no wear, how can the results obtained be related to an actual industrial machining process? Could the average tool wear present in a tool modify the results obtained in the paper concerning the consecutive passes? Maybe make the need or more or less passes to remove the effect from the ceramic milling? Would it make sense a future working line to identify the tool wear range that keeps these results consistent along a time lengthy machining process?

Action: Good comment and this are actually part of the work we are doing right at the moment. However, its resulting in extensive amounts of data and therefore we need to establish this groundwork first before taking the next step of adding the tool wear into the equation. This comment has been added to the discussion/analysing part of the manuscript.

Reviewer 3 Report

Please, you work in a ledaing centre and companies, please do all the suggestions, JMMP must reach a high impact in few years. You can do better.

Author Response

In some place the work looks like a report, but with useful ideas. With some little effort it could

reach the state of a good paper, following quality standards from ASME, as example.

  1. The topic is quite interesting and the results of direct application. In USA is very common

the In718 even Waspaloy and Udimet, and in Europe it seems the same. So I do not

understand why there was not any reference to recent works by Polvorosa and Suarez,

regarding exactly residual stresses and fatigue results. I think one of the authors is the

same. J Manuf Processes is somewhat missed along with ASME Trans, and some MDPI

works as well. Instead the authors get back to old works outdated by the others. Let me

check the ideas: Ezugwu work was old with respect results by Suarez.

Action: Good comments and I have myself spent a lot of my research digging into this. This is the third paper in a series where I want to describe how to determine the machining allowance based on both residual stresses and deformation. This is the last fundamental pieces in that work and I’m preparing a final paper on this to capture all aspects. I agree on the reference side and have been working together with Suárez in the past and hold his work high. I have added some more references but in principle, there are not much done on milling of superalloys, the majority is done for turning.  

ï‚· Fatigue is key in gas turbines, in energy production usually are reported some issues

concerning it

  1. Figure 1 is…ugly the best to say.

Action: I agree, this image has been deleted.

  1. Wretlan work are well known in the business, how did you eliminate some of them.

Action: Mr. Wretland has been a major part of my work as well, he’s one of the co-authors.

  1. Cutting tool angles: rake and relief angles are complex in ceramic rounded tools,

perhaps a drawing including the right position of the very aggressive ceramic tool can

be added up. Is it a commercial toolholder.

ï‚· I suppose they are Sanvik, no Greenleaf?

Action: Additional information has been added to the text regarding the tool geometries and tool holders.

  1. Residual stresses are always a complex matter, Y should prefer hole drilling ASTM test

instead of diffraction XR. X-ray diffraction, using a XStress 3000 G2R, Ok, but how did

you calibrate it?

Action: The method I work with is accredited equipment here at RISE IVF which follows the standards. We perform yearly audits as well as performing references measurements regularly, weekly, on certified reference samples.

ï‚· Statistical analysis is always a complex problem.

  1. Surfaces are under tensile positive stresses. Please discuss it, using missed works, just

on the contrary a complex fatigue campaign would be absolutely necessary. However I

found this Journal of Materials Engineering and Performance 25 (11), 5076-5086 and it

explained a good idea. And Materials and Manufacturing Processes 32 (6), 678-686

Action: Good idea and comments. These are two good publications explaining a lot. I was also a part of the that work but these recommended papers mainly focus on the tool wear influences. Such work is very important and the next natural step in this research as well. I agree that fatigue is the most important functional aspect. However, that is not the focus of this work, instead I would like to establish a methodology of how to define the machining allowance but the next step would be to test the methodology in fatigue perspective as well. I agree that future work requires fatigue testing but as you are aware, this is very time consuming and costly. Hence, it’s important to establish good methods to select robust strategies to machine.  I have added a section in the discussion part addressing the need for future research on testing the methodology in relation to fatigue.

  1. Ceramic up-milling results in the lowest tensile surface stresses while the depth impact

is lesser for down milling and in this perspective, down milling is preferable.

Really, with your own background the paper could scale to one of the good ones, meanwhile

now the references are brochures, old works, and commercial guides. Sandvik is well-known,

but it change every 2-3 years.

Did you think to use cryoMQL, with CO2 or LN2? Some result were promising, but no results of

residual stresses.

Action: The main scope is to develop methodology on incorporating residual stresses into the decision making of allowance. Such work has not been presented before and hence the difficulties in finding good references and the only ones are more industrial since they are the once dealing with this issue. The use of different cooling is very interesting and could be a next step where this methodology is applied.

In brief and plain language: please make a better version, update the ideas and state of the art.

Round 2

Reviewer 3 Report

I think the paper is partial, they did not acknowledge the state of the art properly, and they did not make a discussion. JMMP can be in the future one reference journals, but that obliges in the main continents, America, Asia and Europe to work hard on that aspect that make a work an outstanding work.

I do not agree with answers, but I align myself with previous comment by reviewers.

Some points in addition:

Statistical analysis is the difference between a conference work and journals, How many pieces did you check, are the values the average ones? This is a very weak point. Some authors established comparison with previous works.

All tool definition is weak: how much is exactly the edge roundness, how many tools did you measure.

Ceramic dry milling: all people are concerned about temperature, ceramics inserts are too aggressive, so the following point would be discussed in depth:

  • Temperature measurement, make some comments
  • Thermal relaxation after milling
  • Upmilling in comparison with downmilling: this is key, it could include a new section.

The paper could be for a conference but it is far from being good enough for a conference. People are using Greenleaf tool already in market.

I regret to inform and the paper does not seem in a good state: literature review, results, comparisons…not in a quality good level.

Author Response

Reviewer 3 round 2:

  1. I think the paper is partial, they did not acknowledge the state of the art properly, and they did not make a discussion. JMMP can be in the future one reference journals, but that obliges in the main continents, America, Asia and Europe to work hard on that aspect that make a work an outstanding work.

Response:

It is not our intention at all to be partial. This study is based on the materials, cutting tools and machining parameters used in production of aero engine gas turbines at a manufacturer. We did not work together with any tool manufacturer.

Thank you for these comments and pushing us to make the manuscript even stronger. We have revised the manuscript again focusing on making the scope and focus of this manuscript clearer. In prior, works we have discussed the machining aspects more in detail, but in this work, we intend to focus mainly on the machining allowance assessment methodology. Regarding state of art, we have added new references and broadened the state of art including more aspects of machining that affect the induced stresses. Additionally, the discussion section has been updated as well to put our results in relation to the existing literature.

  1. Statistical analysis is the difference between a conference work and journals, How many pieces did you check, are the values the average ones? This is a very weak point. Some authors established comparison with previous works.

Response:

The results were retrieved from one test piece only, but we performed machining and checked results for two test pieces, showing similar results. I agree that statistical aspects are important but also very difficult. For once, as I mentioned in the added reference the main source of variation comes from the material itself. Additionally, machining setting and tool geometry such as the edge radii, that you accurately point out, may redistribute the stresses and since the cutting edge geometry may vary between different tools as well as with tool wear I expect that this may contribute to more variation. I agree that this would be very interesting to study but it would require a separate study. In fact, we performed machining test on four different test pieces, with worn ceramic inserts as well, and we plan to publish those results in a preceding manuscript discussing the impact from cutting edge, friction and thermal aspects as you suggest.

  1. All tool definition is weak: how much is exactly the edge roundness, how many tools did you measure.

Respones:

We only have access to the used tool and inserts hence, these were measured. All cutting edges were measured and with the averaged results as presented in the manuscript.

  1. Ceramic dry milling: all people are concerned about temperature, ceramics inserts are too aggressive, so the following point would be discussed in depth:
  • Temperature measurement, make some comments
  • Thermal relaxation after milling
  • Upmilling in comparison with downmilling: this is key, it could include a new section.

Respones:

This has been thoroughly discussed in prior work, which is referred to, but we have also added references and a part in the discussion acknowledging this.